# Peer review of "Inflammatory Stimuli Responsive Non-Faradaic, Ultrasensitive Combinatorial Electrochemical Urine Biosensor"

_sensors, 2022, doi:10.3390/s22207757_

Round 1

Reviewer 1 Report

The manuscript showed an interesting study on the development of a sensor for inflammatory biomarkers in combination with statistical treatment of the data. The discussion is adequate and well developed. The only suggestion to authors is to introduce some EIS chart in the manuscript. In this way, we would have a better idea of the impedimetric response of the biosensor.

Author Response

We would like to thank the reviewer for this comment. To add clarity on the impedimetric response, characteristic Bode magnitude and Bode phase plots for IL-8 and IL-6 have been added to the Supplementary information.

Reviewer 2 Report

1. In the work, it is necessary to point out the relevance of modeling the reaction of the biosensor in order to save reagents during its development.

2. The mathematical model used in COMSOL Multiphysics is not presented.

3. To supplement the matrix of inconsistencies of the disease state, it is necessary to conduct an ROC analysis.

Author Response

  1. In the work, it is necessary to point out the relevance of modeling the reaction of the biosensor in order to save reagents during its development.

Response 1:  We agree with the reviewer. The following sentences have been added to the revised manuscript (Page 6, line 231):

“The biosensor operates by transducing subtle variations in the impedance at the electrical double layer (using Electrochemical Impedance Spectroscopy) due to the binding between the target cytokine (expressed picomolar concentration in urine) and its corresponding monoclonal antibody capture probe. Thus, it is important to model and optimize the electrical field behavior of the electrode system before performing the actual biosensing experiments to ensure suitable sensor calibration and to avoid denaturation of the biomolecules due to improper application of voltage (or electrical field). Also, since the sensor uses a combination of electrochemical analysis and lateral flow operation, the fluid field behavior was also studied to assess the operating conditions which was important as it reduced the iterative steps during assay development.”

  1. The mathematical model used in COMSOL Multiphysics is not presented.

Response 2:  The mathematical model and governing equations have been added to the revised supplementary information.

  1. To supplement the matrix of inconsistencies of the disease state, it is necessary to conduct an ROC analysis.

Response 3:  Sklearn library was used to evaluate the Area under Receiver Operator Characteristics curve. These has been added to the revised supplementary information.

Reviewer 3 Report

This is a very nice paper and I have only a few comments.

line 152: "A backing made of plastic adhesive was used to structurally support the sensor. "

How does it look like? What are the dimensions the size of electrodes and so on.

line 184: "The geometric models for analysis were designed using the AutoDesk™ AutoCAD software."
The same as before: that is very nice but (the result) can not be seen in the manuscript. It is also not clear how the flow was estabilished (for which the COMSOL simulation was done).

A figure is really needed that shows how the device looks like in working state and at least the size of the eletrodes should be given on the design or in the text (or figure caption).

There is a mistype (order of words) that should be corrected :
line 232: low very volumes

There is a paper about body fluid (urine) analysis with microelectrodes (a label free technique) that should be cited:

An integrated electro-optical biosensor system for rapid, low-cost detection
of bacteria  (Dániel Petrovszki at al)
https://doi.org/10.1016/j.mee.2021.111523

Author Response

Response to Reviewer Comments

  1. line 152: "A backing made of plastic adhesive was used to structurally support the sensor." How does it look like? What are the dimensions the size of electrodes and so on.

line 184: "The geometric models for analysis were designed using the AutoDesk™ AutoCAD software." The same as before: that is very nice but (the result) can not be seen in the manuscript. It is also not clear how the flow was estabilished (for which the COMSOL simulation was done).

A figure is really needed that shows how the device looks like in working state and at least the size of the eletrodes should be given on the design or in the text (or figure caption).

Response 1:  The details of the electrode system design have been added to the supplementary information (figure S3).

Detailed description of the COMSOL models, and the equations, and the details of fluid flow model optimization have also been added to the supplementary information.

  1. There is a mistype (order of words) that should be corrected: line 232: low very volumes

Response 2:  We thank the reviewer for pointing this out. This has been corrected in the revised manuscript. In addition, the entire manuscript has been thoroughly revised for other typos and grammatical errors.

  1. There is a paper about body fluid (urine) analysis with microelectrodes (a label free technique) that should be cited: An integrated electro-optical biosensor system for rapid, low-cost detection of bacteria (Dániel Petrovszki at al) https://doi.org/10.1016/j.mee.2021.111523

Response 3:  The following reference has been added to the revised manuscript:

Petrovszki, Dániel, Sándor Valkai, Evelin Gora, Martin Tanner, Anita Bányai, Péter Fürjes, and András Dér. “An Integrated Electro-Optical Biosensor System for Rapid, Low-Cost Detection of Bacteria.” Microelectronic Engineering 239–240 (February 15, 2021): 111523. https://doi.org/10.1016/J.MEE.2021.111523.

Reviewer 4 Report

The presented paper fulfills the Journal Scopus. This article is based on 28 articles in the literature over the last fifteen years. The results are clearly stated. The authors, motivated by cited literature, carried out experiments and confirmed known knowledge. The work presents new possibilities for obtaining sensor materials on flexible substrates. The results presented are part of the development of the discipline.

Recommendation Regarding This Manuscript: Accept in the present form

Author Response

We thank the reviewer for taking the time to review the manuscript and for the feedback.